# Functional Study of *PgHDZ01* Gene Involved in the Regulation of Ginsenoside Biosynthesis in *Panax ginseng*

**DOI:** 10.3390/plants14233562

**Published:** 2025-11-21

**Authors:** Chaofan Wang, Xin Cheng, Li Li, Ruicen Liu, Dinghui Wang, Yue Jiang, Yanfang Wang, Kangyu Wang, Mingzhu Zhao, Yi Wang, Meiping Zhang

**Affiliations:** 1College of Life Science, Jilin Agricultural University, Changchun 130118, China; wcftx99@163.com (C.W.); 15568797613@163.com (X.C.); lili@xjau.edu.cn (L.L.); liuruicen777@163.com (R.L.); camellia_wdh@163.com (D.W.); jiangyue285431@163.com (Y.J.); kangyu.wang@jlau.edu.cn (K.W.); zhaomingzhu0125@163.com (M.Z.); 2Research Center for Ginseng Genetic Resources Development and Utilization, Jilin Agricultural University, Changchun 130118, China; yfwang2014@163.com; 3College of Chinese Medicinal Materials, Jilin Agricultural University, Changchun 130118, China

**Keywords:** *PgHDZ01* gene, association analysis, virus-induced gene silencing, functional validation, ginsenoside biosynthesis

## Abstract

The *HD-Zip* gene family plays a vital regulatory role in plant growth and development, but its function in plant secondary metabolism remains to be further elucidated. This study therefore investigated the involvement of the *HD-Zip* gene family in ginsenoside biosynthesis and identified the genes that were highly correlated. Through correlation analysis between gene expression and ginsenoside content, heatmap and co-expression network analyses, and association analysis between Single Nucleotide Polymorphism (SNP) variations and ginsenoside content, the *PgHDZ01* gene was identified as the core candidate gene highly associated with ginsenoside biosynthesis. Subsequently, this gene was shown to significantly respond to Methyl Jasmonate (MeJA) treatment and correlated with changes in ginsenoside content. Further functional verification using Virus-Induced Gene Silencing (VIGS) in *Panax ginseng* (Ginseng) seedlings and gene overexpression in ginseng hairy roots confirmed the positive regulatory role of *PgHDZ01* in ginsenoside biosynthesis. Moreover, its overexpression led to a significant increase in the content of ginsenoside Rb1, and its expression was positively correlated with the change in Rb1 content. The results of this study provide a reference for the comprehensive analysis of the functions of the *HD-Zip* gene family and the regulatory mechanisms of ginsenoside synthesis. The VIGS method used in this study also provides a new approach for the verification of functional genes in ginseng. The results demonstrate the critical role of *PgHDZ01* in regulating ginsenoside biosynthesis, deepening our understanding of the underlying mechanisms of ginsenoside biosynthesis and offering a potential candidate gene for molecular breeding applications.

## 1. Introduction

*Panax ginseng* (Ginseng) is a perennial herb of the *Araliaceae* family, with significant medicinal and economic value. Ginsenosides, its primary active compounds [1], are triterpenoid ginsenosides that can be classified into oleanane-type and dammarane-type ginsenosides. The dammarane-type ginsenosides are further subdivided into the protopanaxadiol-type (Rb1, Rb2, Rb3, etc.) and the protopanaxatriol-type (Re, Rf, Rg1, etc.) [2]. Studies have shown that ginsenosides possess various biological activities, including anticancer [3], anti-inflammatory [4], cardioprotective [5], hepatoprotective [6], and neuroprotective functions [7]. Currently, the biosynthetic pathways of ginsenosides are largely understood, involving the mevalonate (MVA) pathway and the methylerythritol phosphate (MEP) pathway [8], with over 20 key enzyme-coding genes involved [9]. Additionally, some progress has been made in understanding the role of transcription factors in regulating ginsenoside synthesis. For example, in Methyl Jasmonate (MeJA)-treated ginseng callus tissues, the expression of the *bHLH* transcription factor gene showed a positive correlation with the *IspF*, *HMGS*, and *MVK* genes [10]. In ginseng cells, overexpressing the *WRKY* transcription factor gene led to the activation of squalene epoxidase transcription [11]. Furthermore, overexpression of the *NAC* transcription factor gene in ginseng callus tissue results in a significant upregulation of the *DDS* gene, which enhances dammarenediol synthesis [12]. It is noteworthy that existing research on transcription factors has largely focused on the regulation of precursor compound synthesis for ginsenosides, while studies on the regulation of ginsenoside monomer synthesis remain relatively scarce. Therefore, a deeper investigation into the regulatory mechanisms of transcription factors in the synthesis of ginsenoside monomers will not only help in understanding the synthesis process of ginsenosides but also provide a theoretical basis for promoting ginsenoside production.

The homeodomain-leucine zipper (*HD-Zip*) gene family is a plant-specific transcription factor family [13]. The proteins encoded by this family contain both a homeodomain (HD) and a homeobox-associated leucine zipper (HALZ) domain [14]. Based on the conserved domains, the *HD-Zip* gene family is further divided into four subfamilies [15]. This gene family plays a crucial role in plant growth, development, and stress responses. Specifically, *HD-Zip* genes are involved in plant responses to stress factors such as drought [16], salinity [17], and diseases [18], while also regulating the development of plant organs, including roots [19], stems [20], and fruits [21]. In addition, *HD-Zip* genes participate in the regulation of plant hormones, affecting the concentrations of hormones such as gibberellins [22], ethylene [23], and auxins [24], thereby influencing plant growth processes. Recent studies have also suggested that the *HD-Zip* gene family may play a crucial role in the synthesis of secondary metabolites in plants. For instance, in *Vitis vinifera* (Grape), *HD-Zip* genes regulate the synthesis of flavonoids [25]; in *Camellia sinensis* (Tea Plant), they control the synthesis of catechins [26]; and in *Pueraria lobata* (Kudzu), they regulate the synthesis of isoflavones [27]. These findings highlight the important role of *HD-Zip* genes in the synthesis of plant metabolites. However, no studies have been reported on the role of the *HD-Zip* gene family in regulating the biosynthesis of ginsenosides.

Currently, Virus-Induced Gene Silencing (VIGS) has been successfully implemented in plants such as *Arabidopsis thaliana* (Thale Cress) [28], *Glycine max* (Soybean) [29], and *Solanum lycopersicum* (Tomato) [30], but related research in ginseng is still lacking. VIGS is simple to operate, with high silencing efficiency, making it especially suitable for ginseng, a species with difficult cultivation, challenging genetic transformation, and a long growth cycle, providing a rapid approach for gene function validation in this species. However, since ginseng only produces stem and leaf tissues once a year, and the leaves are formed before the growing season, the application of VIGS faces significant challenges.

In our previous studies, the ginseng *HD-ZIP* gene family was identified, comprising 117 family members, and a relatively comprehensive systematic analysis of these genes was conducted. Building on this foundation, the present study aims to employ multiple analytical methods to screen for key genes within this family that are involved in ginsenoside biosynthesis and to further validate their functions. This research will not only help to refine the understanding of *HD-ZIP* gene family functions but also provide a new perspective for elucidating the regulatory mechanisms of ginsenoside biosynthesis, while simultaneously offering potential candidate gene resources for the molecular breeding of *P. ginseng*.

## 2. Materials and Methods

### 2.1. Databases, Plant Material, and Vectors

This study utilized a ginsenoside content database and a transcriptomic database (which includes Single Nucleotide Polymorphism (SNP) mutation data) from 42 farmers’ cultivars of *P. ginseng* in Jilin province [31,32,33]. The identification results, expression data, and materials for MeJA induction experiments related to the *P. ginseng HD-Zip* gene family were derived from previously published research by our laboratory [34,35]. VIGS system-related vectors and strains were provided by the College of Agronomy and the Crop Chemical Control Research Center, China Agricultural University. The pCAMBIA3301 overexpression vector, *Escherichia coli* (*E. coli*) strain DH5α, *Agrobacterium rhizogenes* (*A. rhizogenes*) strain C58C1, and *Agrobacterium tumefaciens* (*A. tumefaciens*) strain GV3101 were preserved in our laboratory. The plant materials used in this experiment included *P. ginseng* seeds preserved in the laboratory and 4-year-old flowering *P. ginseng* plants cultivated at the Jilin Agricultural University plant base. Using *P. ginseng* seeds as the starting material, a unified surface sterilization procedure was applied. The seeds were peeled and rinsed under running water for 30 min, followed by sequential treatment with 75% ethanol (Tianjin Xinbote Chemical Co., Ltd., Tianjin, China) for 1 min and 5% NaClO (Tianjin Xinbote Chemical Co., Ltd., Tianjin, China) for 10 min, and finally rinsed three times with sterile water. Subsequently, three types of explants were prepared via different pathways. The sterilized seeds were subjected to embryo excision and inoculated on 1/2 Murashige and Skoog (MS) medium (Beijing Solarbio Science & Technology Co., Ltd., Beijing, China) for aseptic culture under weak light at 22 °C to obtain aseptic embryos; continuing cultivation for 15 days yielded 15-day-old seedlings. Alternatively, intact sterilized seeds were directly cultured in water to produce hydroponic seedlings. Additionally, healthy leaves from the 4-year-old flowering *P. ginseng* plants were directly used as the experimental material for analysis during their blooming period.

### 2.2. Screening of HD-Zip Genes Highly Associated with Ginsenoside Biosynthesis

First, SPSS (Version 23) software was used to calculate Spearman’s rank correlation coefficient between the expression levels of *HD-Zip* genes and ginsenoside content in Jilin ginseng. A two-tailed test was employed to determine the significance of the correlations. Genes significantly correlated with ginsenoside content in the results were selected as potential candidates (*p*-value ≤ 0.05) for further investigation of their involvement in the regulatory mechanisms of ginsenoside biosynthesis.

Next, to visualize the co-expression patterns between the candidate *HD-Zip* genes and the 16 validated key ginsenoside biosynthesis genes, a heatmap was generated using the pheatmap package in R (version 3.3.3) [36]. Genes exhibiting co-expression trends with key enzymes in ginsenoside biosynthesis were considered potential regulators of ginsenoside synthesis and selected for further analysis. The information for the 16 validated key enzyme genes involved in ginsenoside synthesis is as follows: *PgFPS_22* (DQ087959.1) [37]; *PgSS_1* (AB115496.1) [38]; *PgSE2_1* and *PgSE2_4* (AB003516.1) [39]; *Pgβ-AS_1* and *Pgβ-AS_6* (AB009030.1) [40]; *PgDS_1* and *PgDS_3* (AB122080.1) [41]; *PgCYP716A53v2_1* (JX036031.1) [42]; *PgCYP716A52v2_3* (JX036032.1) [42]; *PgCYP716A47_1* (JN604536.1) [43]; *PgCAS_11, PgCAS_21, PgCAS_22,* and *PgCAS_23* (AB009029.1) [40]; and *PgUGT71A27_2* (KM491309.1) [44].

Simultaneously, the Spearman correlation coefficients between the candidate genes and the 16 key enzyme genes were calculated using R software (version 3.3.3). A two-tailed test was employed to determine the significance of the correlations, with the significance threshold set at *p*-value ≤ 0.05. Co-expression interaction networks were constructed using Bio Layout Express^3D^ (version 3.0) software [45]. *HD-Zip* genes that formed co-expression interaction networks with key enzyme genes were considered closely related to these enzymes and involved in ginsenoside biosynthesis, and thus were selected for further analysis.

Additionally, SNP information for all *HD-Zip* gene transcripts was extracted from the database using Perl (version 5.24.1). The correlation between SNP/InDel mutations in *HD-Zip* genes and ginsenoside content was assessed using *t*-tests and analysis of variance (ANOVA) in SPSS software (Version 23). For *t*-tests, a two-tailed test was performed, with a significance threshold set at *p*-value ≤ 10^−3^. One-way ANOVA was used to evaluate differences between multiple groups, followed by post hoc LSD (Least Significant Difference) test to compare pairwise group differences, with a significance level of *p*-value ≤ 10^−3^. Genes that exhibited a significant association with ginsenoside content changes (*p*-value ≤ 10^−3^) were selected for further analysis, as they may play a critical regulatory role in ginsenoside biosynthesis. Concurrently, the impact of SNP mutations on phenotypic traits was analyzed ([(*the average mono*-*ginsenosides contents of higher group* − *the average mono-ginsenoside contents of lower group*)/*the average mono-ginsenoside contents of lower group* × 100%]) and their locations were determined to be within open reading frames.

Finally, based on the results of the heatmap analysis, network analysis, and the correlation between gene SNP mutations and ginsenoside content, a Venn diagram was constructed to select gene sequences for final functional validation. The multi-step screening pipeline for identifying candidate *HD-Zip* genes is summarized in Table 1.

### 2.3. Functional Validation of PgHDZ01 Gene in MeJA-Induced Ginseng Hairy Roots

MeJA is a critical inducer that modulates secondary metabolite synthesis and gene expression in related pathways [46]. It has been employed in previous studies to identify the genes involved in ginsenoside biosynthesis [47,48]. In the early stage of this study, ginseng hairy root materials were treated with MeJA induction, the ginsenoside content in the hairy roots was measured, and the *CYP* gene was identified as the most suitable internal reference gene for real-time quantitative PCR (qPCR) experiments [34]. Subsequently, using these materials, we examined the relative expression level of *PgHDZ01* under MeJA treatment. RNA was extracted from ginseng hairy roots using the Trizol method, and cDNA was synthesized by reverse transcription using the PrimeScript™ RT Reagent Kit (Takara Biotechnology (China, Dalian) Co., Ltd.). Furthermore, qPCR was performed using the TB Green™ Premix Ex Taq™ kit (Takara Biotechnology (China, Dalian) Co., Ltd.), with the reaction system and time set according to the manufacturer’s instructions. Three biological replicates were designed, and the relative expression level of *PgHDZ01* was calculated using the 2^−∆∆Ct^ method. To preliminarily validate the regulatory role of *PgHDZ01* in ginsenoside biosynthesis, a correlation analysis was performed between the relative expression level of *PgHDZ01* and the ginsenoside content measured in the earlier stage of this study [34]. The correlation was assessed using Spearman’s rank correlation in SPSS software (version 23). A two-tailed test was applied, and a *p*-value ≤ 0.05 was considered statistically significant.

### 2.4. Cloning of PgHDZ01 Gene and Construction of Overexpression Vector

Total RNA was extracted from ginseng samples using the Trizol method and reverse-transcribed into cDNA using Super RT Kit (BioTeke Corporation (China, Wuxi) Co., Ltd.). The target gene was amplified by PCR with specific primers, cloned into the pGM-T vector, and transformed into *E. coli* cells for sequencing to verify the correct sequence. The confirmed fragment was then digested with *XmaI* (New England Biolabs (China, Beijing) Ltd.) and ligated into the similarly digested pCAMBIA3301 expression vector. The recombinant plasmid was finally transformed into *E. coli* competent cells for amplification and verification via colony PCR and plasmid digestion. Once the results confirmed that the fragment lengths of the target gene and the expression vector matched, the recombinant expression vector construction was completed. Finally, the recombinant vector was transformed into *A. rhizogenes* C58C1 competent cells, and single colonies were selected for PCR verification. The successfully verified colonies were considered as genetically transformed engineering strains and stored for future use. The primers used for gene cloning and quantitative real-time PCR in this study are listed in Appendix A.

### 2.5. Screening of PDS Gene for VIGS Gene Silencing and Vector Construction

The mRNA sequence of the *PDS* (phytoene desaturase) gene from plants was downloaded from NCBI and compared with the transcriptome data of Jilin ginseng. The target gene was selected using the Conserved Domains Search tool from NCBI. The recombinant cloning vector containing the *PgPDS* gene sequence was synthesized by Genewiz Inc. (Suzhou Jinweizhi Biotechnology Co., Ltd., Suzhou, China) and transformed into *E. coli* competent cells. After plasmid extraction, the *PgPDS* gene, *PgHDZ01* gene, and pTRV2 vector were subjected to double digestion using *EcoR*I and *Kpn*I (Takara Biotechnology (China, Dalian) Co., Ltd.), and the target fragments were recovered. The pTRV2 vector was ligated with the two gene fragments, and the ligation mixture was transformed into *E. coli* competent cells for PCR verification. Positive clones were selected, and plasmids were extracted for double-digestion verification. The results confirmed that the fragment lengths of the expected genes and vectors matched, indicating successful construction of the recombinant vector. Finally, the recombinant plasmid, the original plasmid, and control plasmid were introduced into *A. tumefaciens* GV3101 competent cells, followed by PCR detection, and the positive bacterial cultures were retained for future use.

### 2.6. Establishment of Ginseng VIGS System and Validation of PgHDZ01 Gene Function

Firstly, 1 mL of GV3101 culture with target plasmids (pTRV1, pTRV2, and recombinant pTRV2) was added to Luria–Bertani (LB) medium with antibiotics (Kan, Rif, Gen) (Shanghai Beyotime Biotech Inc., Shanghai, China), 10 mM MES (Beijing Solarbio Science & Technology Co., Ltd., Beijing, China), and 20 μM acetosyringone (AS) (Beijing Solarbio Science & Technology Co., Ltd., Beijing, China). The culture grew overnight to an OD600 of 0.45. After centrifugation, the pellet was resuspended in sterile water with 10 mM MES, 20 μM AS, and 10 mM MgCl_2_ (Beijing Solarbio Science & Technology Co., Ltd., Beijing, China), and incubated at room temperature for 3 h. Subsequently, various ginseng explants, including aseptic embryos, hydroponic seedlings, and leaves from both 15-day-old seedlings and 4-year-old flowering plants, were subjected to *Agrobacterium*-mediated transformation. The infection methods were tailored to the explant type, including immersion, cotyledon injection, leaf injection, and surface inoculation. Transformation efficiency was calculated based on the emergence of photobleaching phenotypes in plants transformed with the recombinant pTRV2-*PgPDS* construct. Chlorophyll content and the relative expression level of the *PgPDS* gene were measured to validate the silencing efficiency, following the methods detailed in Section 2.3 and the study in [49]. Statistical analysis was performed using SPSS software (version 23). Differences in chlorophyll content between the wild-type and transformed plants were evaluated using a two-tailed independent *t*-test. A *p*-value ≤ 0.05 was considered statistically significant.

After establishing the ginseng VIGS system, it was used to verify the function of the *PgHDZ01* gene. During the verification process, the bacterial suspension prepared in Section 2.5 was used to transform ginseng materials following the method described in Section 2.6. Subsequently, the successfully transformed ginseng leaf samples were used to detect gene expression levels using the method outlined in Section 2.3.

### 2.7. Genetic Transformation for Overexpression of the PgHDZ01 Gene

After silencing the *PgHDZ01* gene through VIGS and validating its function, the *PgHDZ01* gene was subjected to overexpression functional validation. First, healthy ginseng seedlings were cut into 1 cm segments and placed on MS medium containing hormones for pre-culture at 23 °C for 2 days. Then, 1 mL of *A. rhizogenes* C58C1 bacterial culture containing the target plasmid was added to liquid LB medium containing Kan and Rif antibiotics, and cultured overnight to an OD600 of 0.4–0.6. After centrifugation (6000 rpm, 5 min), the supernatant was discarded and the pellet was resuspended in MS medium (Beijing Solarbio Science & Technology Co., Ltd., Beijing, China) containing 20 μM AS. The OD600 was adjusted to 0.4–0.5, and the mixture was activated in a shaking incubator at 28 °C for 2 h. The pre-cultured materials were immersed in the aforementioned resuspension for 10 min for infection. After removing the excess bacterial suspension, the materials were transferred to co-cultivation medium (1/2 MS) containing 20 μM AS and co-cultured for 2 days. After co-cultivation, the materials were transferred to a medium containing cephalexin (Shanghai Beyotime Biotech Inc., Shanghai, China) to induce hairy root formation. Finally, the hairy roots were cut from the explants and placed on solid 1/2 MS medium for further amplification, with each single root numbered sequentially (root 1, root 2, etc.).

### 2.8. Detection of Positive Ginseng Hairy Root Materials and Analysis of Changes in Gene Expression and Ginsenoside Content

Genomic DNA was extracted from 100 mg of ground ginseng hairy root powder using the Genomic DNA Extraction Kit (BioTeke Corporation (China, Wuxi) Co., Ltd.). Six primers (Appendix A) were designed for PCR detection of positive transformants, with hairy roots transformed with pCAMBIA3301:00 serving as a negative control and *A. rhizogenes* C58C1-induced hairy roots as the wild-type control. RNA was then extracted from the positive hairy roots using the method described in Section 2.3 and reverse-transcribed into cDNA. Quantitative PCR was performed to measure the relative expression of the target gene. Finally, ginsenosides were extracted using the Soxhlet method and quantified via high-performance liquid chromatography (HPLC). The chromatographic conditions and mobile-phase elution conditions were established based on the relevant literature [34].

## 3. Results

### 3.1. Identification of HD-Zip Genes Highly Correlated with Ginsenoside Biosynthesis

First, the correlation between the expression of *HD-Zip* genes and ginsenoside content in ginseng was analyzed using SPSS (Appendix A). The results showed that 41 gene transcripts were significantly correlated with ginsenoside content. Among these, 11 transcripts exhibited a highly significant correlation (*p*-value ≤ 0.01), while the other 30 were significantly correlated (*p*-value ≤ 0.05). These transcripts were selected as candidate genes for further analysis. These genes are likely involved in the regulation of ginsenoside biosynthesis and were therefore selected as candidate *HD-Zip* genes for further analysis.

Next, heatmap analysis (Figure 1A,B) indicated that the expression of several candidate *HD-Zip* genes clustered with ginsenoside biosynthesis key genes, including *PgHDZ01*, *PgHDZ18-04*, and *PgHDZ19-07* from four-year-old ginseng in Jilin, and *PgHDZ01*, *PgHDZ26-03*, and *PgHDZ23-01* from 14 tissue types. The five genes exhibited co-expression patterns with ginsenoside biosynthesis genes, indicating a high correlation with ginsenoside biosynthesis. Therefore, these genes were selected for the final Venn diagram analysis.

Simultaneously, the interaction network analysis (Figure 2A,B) showed that in 14 tissue types, when *p*-value ≤ 10^−3^, *PgHDZ01* and *PgHDZ27-04* formed a co-expression network with seven key enzyme genes for ginsenoside biosynthesis. It is noteworthy that when the *p*-value ≤ 10^−4^, only the *PgHDZ01* gene formed a co-expression network with the five key genes involved in ginsenoside biosynthesis, indicating the gene’s significance in ginsenoside synthesis. Moreover, to assess whether the formation of the co-expression network was a random event, 38 sequences were randomly selected from the transcriptome database. This process was repeated 20 times to construct co-expression networks, and statistical analysis was performed on the number of nodes and edges in these networks. The results indicate that the network constructed from *HD-ZIP* candidate genes and ginsenoside biosynthesis key genes exhibits higher compactness compared to the co-expression network generated from random sequences. This suggests that the interaction network formed in this study is not random (Figure 2C–F). Based on the above analysis, *PgHDZ01* and *PgHDZ27-04* were selected for subsequent Venn diagram analysis to further identify candidate genes.

Additionally, further analysis of the 367 SNP mutations across all 117 *HD-Zip* gene transcripts and their correlation with ginsenoside content (Appendix A) revealed that 13 SNPs from 6 transcripts (*PgHDZ01*, *PgHDZ09*, *PgHDZ11*, *PgHDZ12-03*, *PgHDZ13-07*, and *PgHDZ17-05*) were significantly associated with changes in monomer or total ginsenoside content (*p*-value ≤ 10^−3^), suggesting that these genes may play an important regulatory role in the biosynthesis of ginsenosides in ginseng. These six genes were chosen for the final Venn diagram analysis.

Finally, a Venn diagram (Figure 3) was constructed by integrating results from heatmap analysis, network analysis, and correlations between gene SNP mutations and ginsenoside content. Only the *PgHDZ01* gene demonstrated highly significant associations with ginsenoside biosynthesis across all three analyses (combined *p*-value ≤ 10^−9^), strongly suggesting its regulatory role in ginsenoside biosynthesis in *Panax ginseng*. Based on these findings, *PgHDZ01* was selected as the target gene for subsequent functional validation.

### 3.2. Investigation of PgHDZ01 Gene Function in MeJA-Induced Ginseng Hairy Roots

The *PgHDZ01* gene expression in ginseng hairy roots after MeJA treatment is shown in Figure 4. The results indicated that the relative expression level of the *PgHDZ01* gene significantly changed under MeJA induction. Specifically, the gene expression level significantly increased at 6 and 48 h after induction. However, it significantly decreased at 96 and 120 h, suggesting that the *PgHDZ01* gene responds to the regulatory effect of MeJA. This response primarily occurs in the early stages of induction. Additionally, correlation analysis between the previously measured ginsenoside content [34] and *PgHDZ01* gene expression showed a significant correlation with ginsenosides Re and Rf. An extremely significant correlation was also observed with Rb1, Rh1, Rc, Rb2, Rd, F1, F2, and total ginsenoside content (Table 2). These results provide preliminary evidence that the *PgHDZ01* gene plays a role in regulating ginsenoside biosynthesis in ginseng.

### 3.3. Cloning and Overexpression Vector Construction of PgHDZ01 Gene

The full length of the *PgHDZ01* gene is 1485 bp, with an open reading frame (ORF) of 819 bp. The ORF of the *PgHDZ01* gene was cloned using specific primers, as shown in Appendix A. The resulting band was clear and matched the expected length. The *PgHDZ01* fragment used in the VIGS experiment was 517 bp, and the cloning result is shown in Appendix A, with the length consistent with the target sequence. The PCR product was gel-purified, ligated into a cloning vector, and sequenced. A recombinant cloning vector containing the *PgHDZ01* gene was successfully obtained and named pGM-T: *PgHDZ01* and pGM-T: *PgHDZ01*-VIGS. The pGM-T: *PgHDZ01* vector was then digested with *Xma*I, recovered, and ligated into the overexpression vector pCAMBIA3301. The recombinant plasmid was introduced into competent cells, and positive clones were selected and verified via restriction enzyme digestion (Appendix A). The successfully constructed recombinant plasmid was named pCAMBIA3301: *PgHDZ01*. Finally, the plasmid was transformed into *A. rhizogenes* C58C1 competent cells and PCR verification was performed. The positive bacterial culture broth was used in subsequent genetic transformation experiments.

### 3.4. Establishment of the Ginseng VIGS System and Functional Verification of PgHDZ01 Gene

A total of 460 mRNA sequences of the plant *PDS* gene were downloaded from NCBI and aligned with the transcriptome of Jilin ginseng (*p*-value ≤ 10^−6^), resulting in 39 sequences. The Conserved Domains Search confirmed that two transcripts of one gene contained the complete phytoene desaturase domain (PLN02612). ORF Finder and BLAST (Version 2.6.0) alignment showed that the open reading frames (ORFs) of both transcripts were consistent, identifying the gene as ginseng *PgPDS*, with an ORF length of 1746 bp. A 443 bp fragment (479–921 bp) of this gene was selected, synthesized by Genewiz Inc., and used to construct the pGM-T: *PgPDS* recombinant vector. The vector was digested with two restriction enzymes along with pGM-T: *PgHDZ01*-VIGS, ligated, and transformed into competent cells. Positive clones were selected via PCR and further verified with double digestion (Appendix A). The successfully constructed plasmids were named pTRV2: *PgPDS* and pTRV2: *PgHDZ01*-VIGS. Finally, the plasmids were transformed into *A. tumefaciens* GV3101 and PCR verification was conducted. Positive bacterial cultures were used for subsequent genetic transformation experiments.

Subsequently, this study attempted genetic transformation using different ginseng materials. Neither the hydroponic ginseng embryos nor the 15-day-old seedlings were successfully transformed through leaf injection or surface inoculation. However, successful transformation was achieved using bacterial suspension immersion of aseptic ginseng embryos and leaf injection infection of four-year-old flowering ginseng plants. Thirty days later, the aseptic ginseng embryo seedlings immersed in bacterial suspension solution showed a mottled albino phenotype on the leaves and petioles (Figure 5A), confirming that they can be used for VIGS. The VIGS effect was validated by detecting *PgPDS* gene expression (Figure 5B) and chlorophyll content (Figure 5C). In plants with silenced *PgPDS*, *PgPDS* expression was significantly reduced, and the contents of chlorophyll *a*, *b*, and total chlorophyll were significantly lower, with chlorophyll *b* decreasing by approximately 75%. Furthermore, pTRV2: *PDS* injection into the four-year-old ginseng leaves also resulted in a white phenotype in the fruit, demonstrating that VIGS injection could silence genes in the fruit (Figure 5D). These results demonstrate that a VIGS system has been successfully established in *Panax ginseng*, which is applicable across its different growth stages.

Thereafter, utilizing the established VIGS system, the *PgHDZ01* gene was silenced using pTRV2: *PDS* and pTRV2: 00 as positive and negative controls, respectively. After the positive control plants exhibited the albino phenotype, the expression of *PgHDZ01* and four related ginsenoside biosynthesis genes was analyzed (Figure 6). The results showed that in the *PgHDZ01*-silenced plants, *PgHDZ01* gene expression was significantly reduced, and the expressions of four ginsenoside biosynthesis key genes were significantly lower, especially *PgDDS* and *UGT71A27*, which decreased by more than 50%. These findings suggest that the *PgHDZ01* gene plays a positive regulatory role in ginsenoside biosynthesis and may participate in the synthesis of ginseng ginsenosides by regulating the expression of key enzyme genes.

### 3.5. Genetic Transformation of Overexpression Vector and Detection of Positive Transformants

The transformation of aseptic ginseng seedlings was carried out using the C58C1 engineered strain containing the recombinant plasmid pCAMBIA3301: *PgHDZ01*. The transformation process included cutting the aseptic seedlings, infecting them, and performing induction culture, finally obtaining 124 single-root hairy root lines (Figure 7). After screening, 2 hairy root lines transformed with pCAMBIA3301: 00 and 18 transformed with pCAMBIA3301: *PgHDZ01* were obtained. The PCR test (Appendix A) showed that 2 hairy root lines transformed with pCAMBIA3301: 00 were successfully transformed, and 14 transformed with pCAMBIA3301: *PgHDZ01* were positive, which were lines No. 3, 27, 62, 63, 64, 78, 54, 61, 65, 67, 83, 86, 87, and 107. Among them, hairy root lines 61, 83, and 87 had incomplete vector sequences, possibly due to gene recombination. Ultimately, two negative and seven positive hairy root lines that met the suspension culture requirements were cultured, and their ginsenoside content was measured.

### 3.6. qPCR and Ginsenoside Content Analysis of Positive Hairy Roots

The relative expression levels of the *PgHDZ01* gene in the seven positive single-root hairy root lines were analyzed using qPCR (Figure 8). The results showed that the expression of the *PgHDZ01* gene in all positive hairy root lines was significantly higher than in the negative control, with hairy root line 67 showing the smallest increase in expression (1.15-fold) and hairy root line 54 showing the largest increase (7.26-fold). This indicates that the *PgHDZ01* gene was successfully overexpressed in ginseng hairy roots. HPLC analysis quantified nine ginsenosides (Rb1, Rc, Rb2, Rd, Rg3, Rh2, Rf, Rg1, Re) in both the negative control and transgenic hairy root lines, with all contents expressed as milligrams per gram dry weight (mg/g DW, where DW: dry weight) (Figure 9). Compared to the negative control (Figure 9), the content of Rb1 was significantly increased in five of the positive hairy root lines overexpressing the *PgHDZ01* gene, while the content of Rg1, Rg3, Re, and Rd was significantly reduced in four of the hairy root lines. These results suggest that the *PgHDZ01* gene plays a significant regulatory role in ginsenoside biosynthesis. Moreover, SPSS analysis revealed a significant correlation between the relative expression levels of *PgHDZ01* and Rb1 content (Table 3), suggesting that the *PgHDZ01* gene may specifically regulate the biosynthesis of ginsenoside Rb1.

## 4. Discussion

The ginsenoside content in *P*. *ginseng* is a quantitative trait, influenced by multiple genes, making the identification of regulatory genes challenging. For instance, traditional map-based cloning techniques face limitations in the application to quantitative trait genes, as overlapping effects between genes may reduce the precision of localization, thereby restricting their use [50]. Additionally, the phenotypes of ginsenosides are highly complex, with the possibility of interconversion between different ginsenoside monomers [51,52]. This complicates the direct validation of regulatory genes using phenotypic data. Furthermore, transcription factors often regulate multiple target genes, which may interact with one another, adding further complexity to the regulatory process. These factors collectively make it difficult to carry out research on the regulation of ginsenoside biosynthesis. With the rapid advancement of omics technologies, several new methods for identifying functional genes have emerged, such as expression Quantitative Trait Locus (eQTL) and genome-wide association studies (GWAS) [53,54]. These techniques enable large-scale and rapid identification of potential functional genes. However, they often lack specificity, typically identifying dozens or even hundreds of candidate genes [55,56,57]. This creates difficulties for subsequent gene function validation and mechanistic analysis. Given the theoretical challenges in the regulatory mechanisms of ginsenoside biosynthesis and the foundational difficulties in functional gene identification, our laboratory developed a novel method in previous studies for the efficient and accurate identification of genes associated with ginsenoside biosynthesis. This method integrates multiple screening strategies, including correlation analysis of ginsenoside content, heatmap analysis, interaction network analysis, and SNP analysis. The design of these strategies was based on the principle that genes controlling the same trait tend to form interaction networks [58], complemented by molecular marker analysis of quantitative trait loci (QTL) maps [59] and GWAS [60]. This multi-faceted, integrative approach effectively enhanced the reliability of functional gene identification, surpassing the outcomes achievable through any single analysis method. This method has been successfully applied to the identification of genes such as *CYP*, *NAC*, and *OSC* [32,61,62]. Therefore, in this study, this method was used to identify key functional genes in the *HD-Zip* gene family that regulate ginsenoside biosynthesis, and their functions were validated through genetic transformation experiments. These results further demonstrate the reliability, accuracy, and efficiency of our previous screening method, providing a valuable reference for future applications in functional gene research.

VIGS is a crucial tool for plant gene function analysis. Compared to traditional transgenic methods, VIGS offers significant advantages, particularly in the rapid identification and validation of plant functional genes. It allows for the specific silencing of target gene expression using viral vectors, thereby inhibiting gene function and enabling the observation of phenotypic changes. This technique has been applied in various plant species. For instance, in crops such as tomato and *Capsicum annuum* (Pepper), researchers effectively silenced key genes using the root-dipping method [63]. Additionally, injection and rubbing methods have been employed in other species, such as *Lycoris chinensis* (Chinese Stonegarlic) and *Panicum virgatum* (Switchgrass), to perform VIGS experiments [64,65]. Although VIGS technology has been widely applied in various plant species, it has not yet been established in ginseng, which has restricted the rapid identification and validation of functional genes in this plant. Therefore, this study established a VIGS system for ginseng. During the process, multiple methods were tested, and it was observed that ginseng true leaves exhibited a high sensitivity to VIGS, with wilting and necrosis occurring after infection, similar to the response seen in many other plants [66,67]. However, in other plants, even if the leaves wilt after infection, gene silencing can still be achieved as new leaves grow, without leading to plant death. Consequently, alternative methods were explored, and it was found that infection through ginseng embryo immersion allowed the successful development of the majority of embryos. This approach thus confirmed the feasibility of the VIGS system in ginseng. However, due to the small plant size and low dry matter retention rate, despite the ability to perform large-scale transformation, there were still difficulties in ginsenoside content analysis, which could only be assessed through qPCR to measure gene expression levels.

Additionally, due to the unique growth characteristics of ginseng, selecting the appropriate infection timing presents additional challenges when applying VIGS technology. Specifically, the aboveground parts of ginseng (excluding the fruit) are fully developed by the previous year, while the gene silencing phenotype required for VIGS experiments only appears in newly developed leaves after infection. As a result, mature ginseng plants fail to produce new leaves exhibiting phenotypic changes following VIGS treatment. To address this issue, we attempted leaf transformation experiments before the flowering period of ginseng, which resulted in the induction of a mottled whitening phenotype in immature fruits, further validating the method’s effectiveness. Notably, a decrease in fruit number was observed following VIGS treatment. Although the injection of *Agrobacterium* solution caused leaf wilting and resulted in a relatively low overall transformation efficiency, the reduced fruit set rate may be attributed to the toxic effects of the bacterial solution on flower organ development. In future studies, the survival rate of transformed materials can be further enhanced by optimizing the infection system, for instance, through the use of specific infiltration buffers, the trialing of less phytotoxic *Agrobacterium* strains, or a combination with other gene silencing techniques for functional verification. This approach will enable the acquisition of a sufficient quantity of genetically silenced plant materials for determining the ginsenoside content. Consequently, a more comprehensive functional characterization of the *PgHDZ01* gene in ginsenoside biosynthesis will be achieved. In addition, the *PgPDS* gene identified in this study was employed for VIGS transformation in *Nicotiana benthamiana* (Tobacco), with reference to the method used in tobacco VIGS experiments [68]. The results showed that the silencing success rate of the ginseng *PgPDS* gene on the *PDS* gene in tobacco was approximately 80%, resulting in a white phenotype in tobacco (Appendix A). However, the similarity between the ginseng *PgPDS* gene and the tobacco *PDS* gene (DQ469932.1) was 84%, suggesting that the similarity requirement for the *PDS* gene sequence in the VIGS system may not be as strict, which could make this system more widely applicable.

MeJA, a lipid-derived signaling molecule found in plants, is commonly used to study and validate genes involved in ginsenoside biosynthesis, as it induces the expression of several key enzyme genes in the biosynthetic pathway [47,69,70]. In this study, preliminary functional validation of *PgHDZ01* was conducted using MeJA treatment. A significant discrepancy was observed between the results from MeJA treatment and those from *PgHDZ01* overexpression. This divergence may be attributed to the ability of MeJA to activate multiple signaling pathways concurrently, potentially inducing coordinated regulation within a broader stress-response network involving *PgHDZ01* and other genes, rather than reflecting a direct regulatory role of *PgHDZ01* over all ginsenoside biosynthesis pathways. In contrast, the subsequent *PgHDZ01* overexpression experiment provided a more direct assessment of its specific function, without the confounding effects of other MeJA-induced signals. Consequently, a discrepancy in ginsenoside content alterations was observed between the MeJA treatment and the *PgHDZ01* overexpression conditions.

Subsequently, upon further analysis of the changes in ginsenoside content following *PgHDZ01* gene overexpression, a significant increase in Rb1 accumulation was observed in five hairy root lines. This suggests that the *PgHDZ01* gene plays a critical role in positively regulating the biosynthesis of Rb1. However, hairy root line 86 exhibited substantial differences in ginsenoside content, with significant reductions in six diol-type ginseng ginsenosides (Rb1, Rc, Rb2, Rd, Rg3, and Rh2) compared to the other hairy root lines. This difference may be attributed to the loss of function of key enzyme genes due to the insertion of exogenous DNA. Although this hypothesis requires further validation, it still provides valuable experimental material for identifying key genes involved in the biosynthesis of diol-type ginsenosides.

Further analysis revealed that the levels of other protopanaxadiol-type ginsenosides (such as Rd and Rg3) and protopanaxatriol-type ginsenosides (such as Rg1 and Re) were significantly reduced in the overexpression lines. It is speculated that this phenomenon may be associated with the specific regulation of certain glycosyltransferases downstream of the ginsenoside biosynthesis pathway by *PgHDZ01*, which in turn alters the direction of metabolic flux. Specifically, based on the affected ginsenoside types and their biosynthetic pathways, two potential regulatory mechanisms may be involved.

First, *PgHDZ01* may alter the metabolic flux by influencing the distribution of ginsenoside precursors toward protopanaxadiol-type ginsenosides. Specifically, *PgHDZ01* may upregulate the expression of key enzyme genes such as *UGTPg1* (KM491309.1) and *UGT74AE2* (JX898529). The enzyme encoded by *UGTPg1* catalyzes the conversion of PPD to CK [71], while that encoded by *UGT74AE2* catalyzes the conversion of PPD to Rh2 [72]. These regulations may promote the conversion of a common precursor to PPD-type ginsenosides, thereby reducing the availability of precursors used for the synthesis of PPT-type ginsenosides (such as Rg1 and Re), ultimately leading to a decrease in PPT-type ginsenoside levels. Second, within the PPD-type ginsenoside biosynthesis pathway, *PgHDZ01* may further influence ginsenoside accumulation by regulating the specific synthesis route of Rb1. Specifically, *PgHDZ01* may upregulate the expression of genes such as *UGTPg1*, which facilitates the conversion of Rg3 to Rd [71], and *UGTPg71A29* (MH638345), which catalyzes the conversion of Rd to Rb1 [73], thereby enhancing the glycosylation reactions from Rg3 to Rd and subsequently to Rb1. This process may reduce the levels of intermediate products, such as Rd and Rg3, while promoting the accumulation of Rb1. In summary, it is hypothesized that *PgHDZ01* may regulate the differential accumulation of various ginsenoside monomers through the two mechanisms outlined above. However, this hypothesis has limitations, as the proposed regulatory network is based on phenotypic observations and known metabolic pathways. The precise molecular mechanisms will require further validation through future experiments, such as transcriptome sequencing and chromatin immunoprecipitation.

## 5. Conclusions

This study integrated various bioinformatics approaches, including correlation analysis of ginsenoside content, heatmap analysis, co-expression network analysis, and SNP association analysis, to identify the candidate gene *PgHDZ01*, which is strongly associated with ginsenoside biosynthesis. Further validation through experiments such as MeJA induction, VIGS, and overexpression revealed that *PgHDZ01* positively regulates ginsenoside biosynthesis, with its expression level significantly correlating with the content of ginsenoside Rb1. These results suggest that *PgHDZ01* plays a key role in the regulation of ginsenoside biosynthesis. Additionally, a VIGS system was successfully established in ginseng, which offers advantages such as ease of material cultivation, convenience of preservation, and independence from seasonal limitations, and allows for large-scale genetic transformation. This system provides a novel approach for the rapid functional validation of ginseng genes. Future studies can focus on *PgHDZ01* as a starting point, exploring its downstream target genes and associated interaction networks to further elucidate the transcriptional regulation mechanisms underlying ginsenoside biosynthesis. This will provide an important theoretical foundation and technical support for research in areas such as molecular breeding and metabolic engineering of ginseng.

## Figures and Tables

**Figure 1 plants-14-03562-f001:**
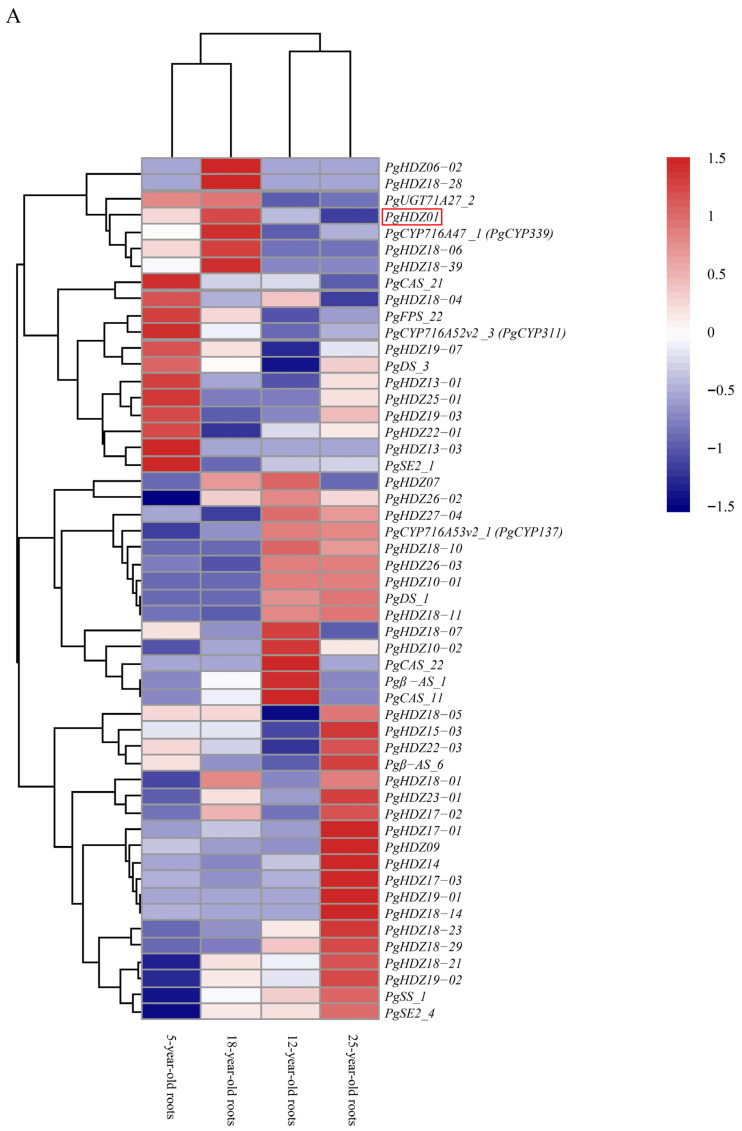
Expression patterns of 41 *PgHD-Zip* candidate genes and key ginsenoside biosynthesis genes in Panax ginseng. The heatmap depicts gene expression levels across (**A**) four growth years (5-, 12-, 18-, and 25-year-old roots) and (**B**) 14 tissue types from a whole 4-year-old plant. Tissue types are fiber root, leg root, main root epidermis, main root cortex, rhizome, arm root, stem, leaf peduncle, leaflet pedicel, leaf blade, fruit peduncle, fruit pedicel, fruit flesh, and seed. The color scale from blue to red indicates low to high expression levels.

**Figure 2 plants-14-03562-f002:**
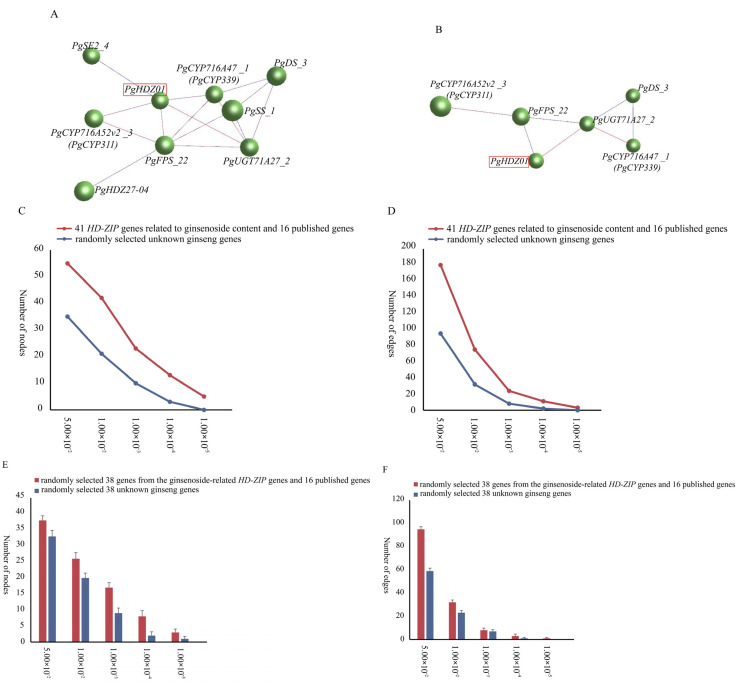
Co-expression networks of 41 *PgHD-Zip* candidate genes associated with ginsenoside content and 16 published ginsenoside biosynthesis genes. The interaction networks were constructed with different *p*-value thresholds: (**A**) ≤10^−3^ and (**B**) ≤10^−4^. Node and edge comparisons were made between the networks of 41 *PgHD-Zip* candidate genes and 16 key ginsenoside biosynthesis genes versus 57 randomly selected genes from the ginseng transcriptome database at various *p*-value thresholds (**C**,**D**). Networks were also constructed by randomly selecting 38 sequences from the 41 *PgHD-Zip* candidate genes, 16 ginsenoside biosynthesis genes, and the ginseng transcriptome database. The number of nodes and edges was calculated for each network at different *p*-value thresholds, with the analysis repeated 20 times for robustness (**E**,**F**).

**Figure 3 plants-14-03562-f003:**
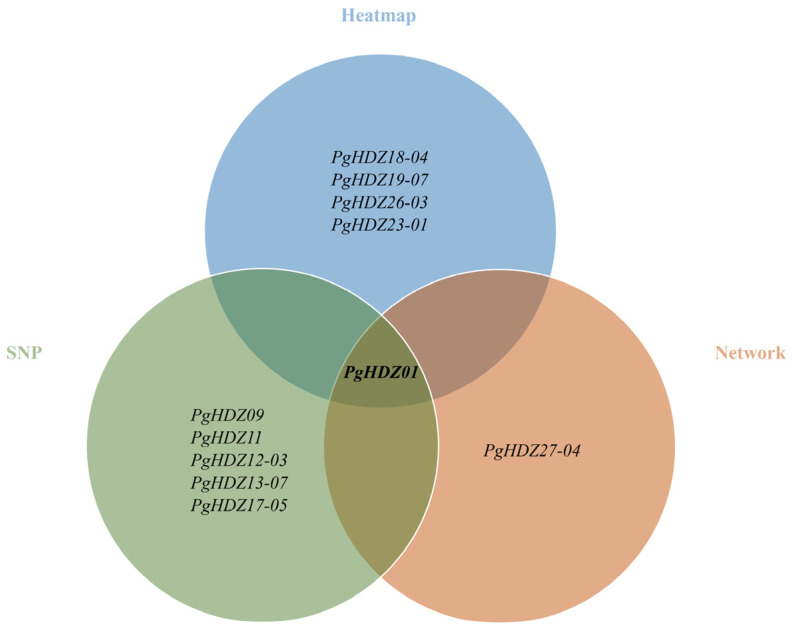
Venn diagram showing the identification of *PgHD-Zip* genes with significant associations with ginsenoside biosynthesis. Heatmap: Clustering results of *PgHD-Zip* genes and 16 key genes involved in ginsenoside biosynthesis, based on their expression patterns across various conditions. Network: Co-expression network results showing the interactions between *PgHD-Zip* genes and the 16 key ginsenoside biosynthesis genes. SNP: Correlation analysis results between mutations in *PgHD-Zip* genes (SNPs) and observed changes in ginsenoside content.

**Figure 4 plants-14-03562-f004:**
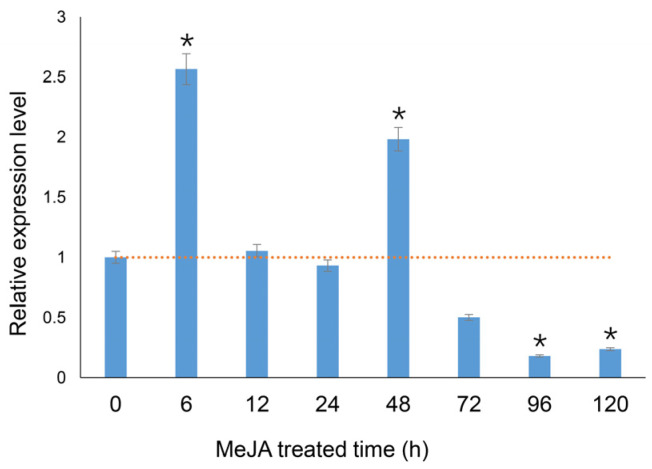
Function validation of the *PgHDZ01* gene, which is highly associated with ginsenoside biosynthesis, in MeJA-induced hairy roots of ginseng. The figure shows the relative expression levels of the *PgHDZ01* gene, which is strongly correlated with ginsenoside synthesis, under MeJA treatment. The *X*-axis represents the treatment duration (0–120 h), while the *Y*-axis shows the relative gene expression levels. The dashed horizontal line indicates the baseline expression level (set as 1.0) of the untreated control (0 h). Asterisks (*) indicate statistically significant differences, with * *p*-value ≤ 0.05.

**Figure 5 plants-14-03562-f005:**
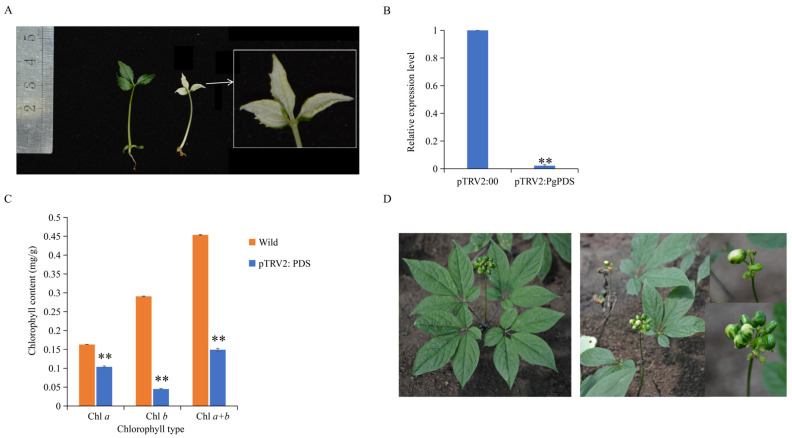
Establishment of the ginseng VIGS system. (**A**) VIGS gene silencing of pTRV2: *PgPDS* using ginseng embryos as explants. The left panel shows wild-type ginseng plants with normal pigmentation, while the right panel shows pTRV2: *PgPDS*-silenced ginseng plants exhibiting albino-like plants. (**B**) Detection of relative expression level of *PgPDS* gene after VIGS. The *X*-axis compares *PgPDS* gene expression levels between the VIGS-silenced pTRV2: 00 control group (empty vector) and albino plants in which the *PgPDS* gene was silenced with pTRV2: *PgPDS*. The *Y*-axis represents relative gene expression levels. Asterisks (*) denote statistically significant differences (** *p*-value ≤ 0.01). (**C**) Comparison of chlorophyll content in wild-type ginseng leaves versus pTRV2: *PgPDS*-silenced ginseng leaves. The *X*-axis represents different chlorophyll types (e.g., chlorophyll *a*, *b*), and the *Y*-axis shows chlorophyll content measured in mg/g of leaf tissue. Asterisks (*) denote statistically significant differences (** *p*-value ≤ 0.01). (**D**) VIGS gene silencing of pTRV2: *PgPDS* using four-year-old flower-stage ginseng leaves as explants. The left panel shows wild-type plants with normal leaf and flower pigmentation, while the right panel shows pTRV2: *PgPDS*-silenced plants with white fruits, indicating successful silencing of *PgPDS*.

**Figure 6 plants-14-03562-f006:**
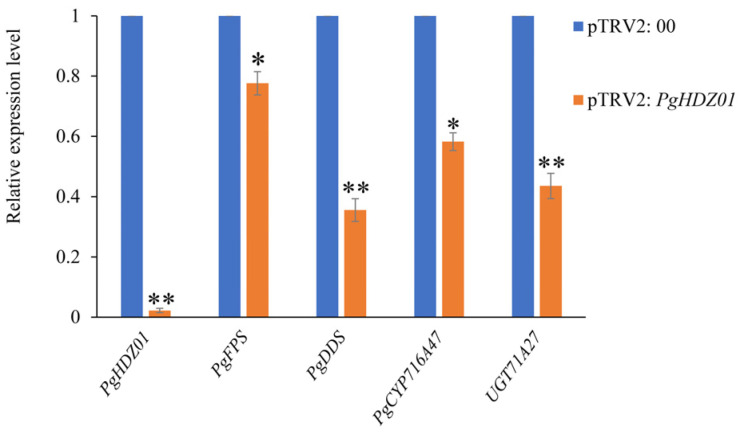
Detection of relative expression levels of ginsenoside biosynthesis-related genes after VIGS of the *PgHDZ01* gene. The *X*-axis shows a comparison of the expression levels of the *PgHDZ01* gene and key genes involved in ginsenoside biosynthesis between two groups: the VIGS-silenced pTRV2: 00 control group and the pTRV2: *PgHDZ01*-silenced albino plants. The *Y*-axis represents the relative gene expression levels. Asterisks (*) indicate statistically significant differences between groups (* *p*-value ≤ 0.05; ** *p*-value ≤ 0.01).

**Figure 7 plants-14-03562-f007:**
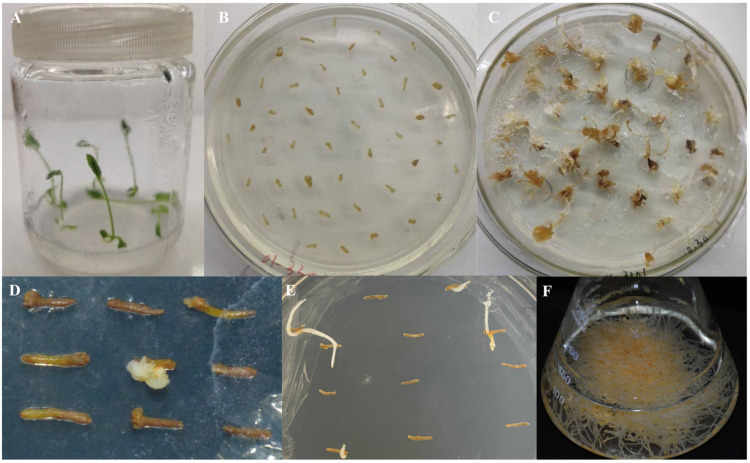
Induction of ginseng hairy roots. (**A**) Ginseng seedlings cultured under standard conditions. (**B**) Various tissue parts of ginseng seedlings, including roots and stems, were cut into small sections for pre-culture. (**C**) Induction culture of ginseng hairy roots. Explants were inoculated with *A. rhizogenes* for co-cultivation, followed by transfer to selection medium. (**D**,**E**) Single-root culture of the induced hairy roots. Induced hairy roots were excised and subcultured individually for further growth. (**F**) Expansion culture of a single induced hairy root line in liquid medium to promote growth.

**Figure 8 plants-14-03562-f008:**
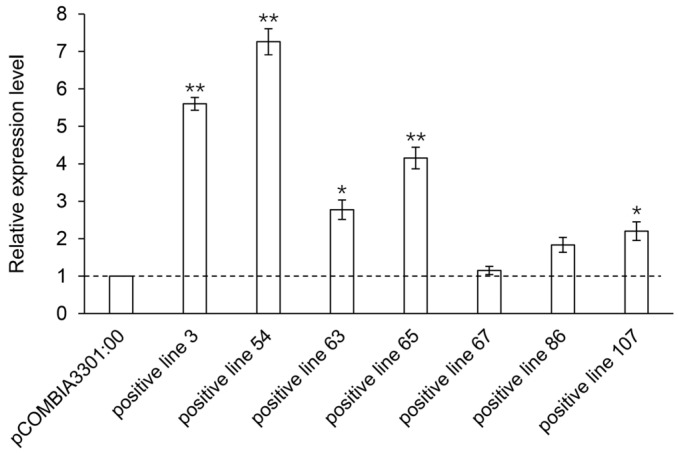
Relative expression levels of the *PgHDZ01* gene in positive hairy root lines. The *X*-axis represents the different ginseng hairy root lines that have been positively overexpressed with *PgHDZ01*, while the *Y*-axis shows the relative gene expression levels measured using quantitative PCR. Asterisks (*) denote statistically significant differences between the overexpression lines and control lines (* *p*-value ≤ 0.05; ** *p*-value ≤ 0.01).

**Figure 9 plants-14-03562-f009:**
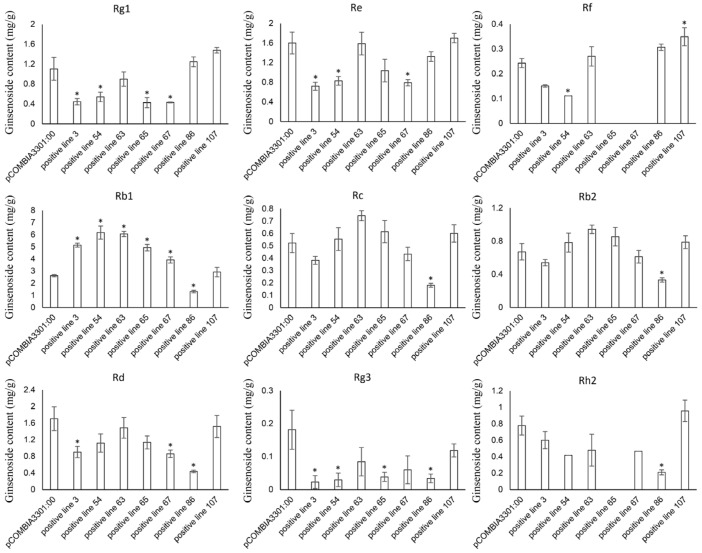
Mono-ginsenoside content in positive hairy root lines that overexpressed *PgHDZ01*. The *X*-axis represents the different ginseng hairy root lines with positive overexpression of *PgHDZ01*, while the *Y*-axis shows the ginsenoside content (mg/g DW) measured from these lines. Asterisks (*) denote statistically significant differences between the overexpression lines and the control group (* *p*-value ≤ 0.05).

**Table 1 plants-14-03562-t001:** This table summarizes the multi-step screening pipeline for identifying candidate *HD-Zip* genes involved in ginsenoside biosynthesis. In Step 1, expression–trait correlation analysis was performed using Spearman’s correlation in SPSS (*p*-value ≤ 0.05) to screen genes correlated with ginsenoside content. Step 2 involved three complementary analyses: (2.1) heatmap clustering analysis to select candidate genes that cluster with key genes involved in ginsenoside biosynthesis; (2.2) co-expression network analysis to select candidate genes that interact with key genes in ginsenoside biosynthesis; and (2.3) SNP mutation analysis to select candidate genes significantly associated with changes in ginsenoside content. Step 3 integrated results from all three methods in Step 2 using Venn diagram to select high-confidence candidate genes for experimental validation.

Screening Step	Method (Software)	Selection Criteria	Objective
1 Trait correlation analysis	Spearman’s correlation (SPSS, Version 23)	*p*-value ≤ 0.05	Screening genes correlated with ginsenoside content
2.1 Heatmap analysis	Heatmap clustering (R, version 3.3.3)	Co-expression trend with ginsenosides biosynthesis key genes	Screening genes highly correlated with ginsenoside biosynthesis
2.2 Network analysis	Co-expression network (Bio Layout^3D^, version 3.0)	Forming interaction networks with ginsenosides biosynthesis key gene	
2.3 SNP mutation analysis	*t*-test/ANOVA (SPSS)	*p*-value ≤ 10^−3^	
3 Integration analysis	Venn diagram (PowerPoint 2016)	Supported by the three methods from Step 2	Selecting high-confidence candidate genes for experimental validation

**Table 2 plants-14-03562-t002:** Correlation analysis of *PgHDZ01* gene expression with ginsenoside content in MeJA-treated ginseng hairy roots. The table presents the statistical relationships between the expression levels of *PgHDZ01* and the content of various ginsenosides in response to MeJA treatment. Asterisks (*) denote statistically significant differences: * *p*-value ≤ 0.05 and ** *p*-value ≤ 0.01.

Ginsenoside	Correlation Coefficient	Sig. (Two-Tailed)
Re	0.543 *	4.49 × 10^−2^
Rf	0.552 *	4.08 × 10^−2^
Rb1	0.800 **	1.99 × 10^−4^
Rg2	0.536	2.15 × 10^−1^
Rh1	0.782 **	3.41 × 10^−4^
Rc	0.744 **	9.48 × 10^−4^
Rb2	0.839 **	6.43 × 10^−4^
Rb3	0.073	8.41 × 10^−1^
F1	0.682 **	3.59 × 10^−3^
Rd	0.764 **	6.23 × 10^−3^
F2	0.755 **	4.51 × 10^−3^
PPT	0.280	3.54 × 10^−1^
Rh2	0.059	8.40 × 10^−1^
PPD	0.459	9.85 × 10^−2^
TOTAL	0.782 **	5.70 × 10^−4^

**Table 3 plants-14-03562-t003:** Correlation analysis of *PgHDZ01* gene expression with mono-ginsenoside content in ginseng hairy roots that overexpress *PgHDZ01*. The table presents the correlation between the expression levels of the *PgHDZ01* gene and the mono-ginsenoside content (mg/g DW) in the overexpressed lines. Asterisks (*) denote statistically significant differences between the overexpression and control groups (* *p*-value ≤ 0.05).

Ginsenoside	Correlation Coefficient	Sig. (Two-Tailed)
Rg1	−0.357	3.85 × 10^−1^
Re	−0.452	2.60 × 10^−1^
Rf	−0.600	2.08 × 10^−1^
Rb1	0.810 *	1.50 × 10^−2^
Rc	0.262	5.31 × 10^−1^
Rb2	0.262	5.31 × 10^−1^
Rd	−0.143	7.36 × 10^−1^
Rg3	−0.548	1.60 × 10^−1^
Rh2	−0.179	7.02 × 10^−1^
TOTAL	0.024	9.55 × 10^−1^

## Data Availability

The data used in this study are available at the Sequence Read Archive (SRA) of the National Center for Biotechnology Information (NCBI) under BioProject PRJNA302556, and at the Gene Expression Omnibus (GEO) of NCBI under SRP066368 and SRR13131364-SRR13131405.

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
