# Peer review of "Functional Study of PgHDZ01 Gene Involved in the Regulation of Ginsenoside Biosynthesis in Panax ginseng"

_plants, 2025, doi:10.3390/plants14233562_

Round 1
Reviewer 1 Report
Comments and Suggestions for Authors
I recommend performing another language review, as some sentences are inappropriate, too long, and unclear, and some terminology is insufficiently professional. Additionally, the authors should pay attention to the article formatting.
I would also like to highlight that the discussion reads more like a summary of the article rather than a focused discussion of the findings. The same applies to the conclusion. Other comments are noted directly in the attached article.

Author Response
Authors’ Responses to Editor and Reviewers
We sincerely thank the editor and reviewers for dedicating a significant amount of time and effort to provide valuable suggestions and feedback for this study. We have carefully reviewed all the comments and suggestions and have provided responses to each reviewer's feedback on the manuscript revision webpage. Additionally, in response to each comment, we have clearly marked the location of every modification in the manuscript with the format "Line: [line number]" at the end of our reply. We hope this precise referencing will facilitate your review process.
Should you need additional information, please do not hesitate to let me know.
With my very best regards.
Chaofan Wang, Ph.D.
Title: Functional Study of PgHDZ01 Gene Involved in the Regulation of Ginsenoside Biosynthesis in Panax ginseng
Manuscript ID: plants-3859842
According to the comments from the editor and reviewers, we have re-uploaded the file "plants-3859842-revisions-1." Additionally, we have also uploaded the file "plants-3859842-revisions-1-with the changes highlighted. In this file, relevant comments regarding the responses to the questions raised by the editor and reviewers have been marked in "Track Changes" mode in Microsoft Word. At the same time, in certain sections where supplementary content was added, we have included the necessary references. The updated references are listed in the References section. In addition to addressing the editor's and reviewers' comments, we also identified some errors during our review of the manuscript and made the necessary corrections. The modified sections are as follows:
- On line 108, information about Agrobacterium tumefaciens ( tumefaciens) GV3101 has been added.
- On line 222, information about Agrobacterium tumefaciens ( tumefaciens) GV3101 has been added.
- On line 398, information about Agrobacterium tumefaciens ( tumefaciens) GV3101 has been added.
- On line 411, "chlorophyll b" has been changed to italicized chlorophyll b.
Editor and Reviewer comments:
Editor comment:
- Please carefully check the accuracy of names and affiliations.
Authors’ responses:
Thanks. We have checked and corrected the formatting of the names and affiliations (Line:3-11).
Reviewer #1 comment:
- The reviewer recommend performing another language review, as some sentences are inappropriate, too long, and unclear, and some terminology is insufficiently professional.
Authors’ responses:
Thank you for your valuable suggestions. We have thoroughly reviewed and revised the manuscript, and have also utilized the professional language editing services provided by MDPI. To confirm the manuscript has been revised according to academic journal standards, we have attached the certificate of language editing. This document verifies that improvements were made to grammar, commonly used terminology, and overall writing quality. The attachment is named English-Editing-Certificate.pdf. We believe these revisions meet the high standards required by the journal, and significantly enhance the readability and professionalism of the manuscript.
- The authors should pay attention to the article formatting.
Authors’ responses:
Thanks. We sincerely apologize for these issues. They likely occurred during the final typesetting or file conversion process. Thank you for bringing them to our attention. We have thoroughly checked the entire manuscript and corrected the errors. All modifications have been made in accordance with the author guidelines. We greatly appreciate your diligence in reviewing our work (Line:6-7; 247; 440-446; 466-485; 687-708).
- I would also like to highlight that the discussion reads more like a summary of the article rather than a focused discussion of the findings.
Authors’ responses:
Thank you very much for your valuable feedback. Based on your suggestion, we have revised and expanded the discussion section to ensure it focuses more on the interpretation of our findings, rather than simply summarizing the article. We believe these changes have improved the clarity and depth of the discussion. We greatly appreciate your thoughtful input, which has helped us enhance the quality of the manuscript (Line:529-662).
- The reviewer recommend that you clarify the objective of your study more explicitly in Abstract.
Authors’ responses:
Thank you very much for your valuable suggestion. In response, we have thoroughly revised the abstract to explicitly and clearly state the objective of our study at the beginning. We believe this modification significantly enhances the clarity of our research purpose and makes it easier for readers to immediately grasp the core of our work (Line:13-31).
- Please revise the keywords to ensure they comply with the journal’s guidelines for keyword formatting.
Authors’ responses:
Thank you for your valuable feedback. We have carefully revised both the content and formatting of the keywords to ensure they comply with the journal’s guidelines (Line:32-33).
- I recommend that the Introduction focus more on what was done in the study rather than on the results.
Authors’ responses:
Thank you for your valuable suggestions. Based on your feedback, we have revised the introduction to focus more on the process, methods, and significance of the study, rather than emphasizing the research results. This revision helps to make the introduction clearer and will assist readers in better understanding the background and objectives of the research, while also improving the overall logical flow of the article (Line:73-76; 85-93).
- Were the plants from different cultivars grown under the same conditions? How was environmental uniformity ensured to minimize its impact on the results?
Authors’ responses:
Thank you for your thoughtful question. Apologies for any previous lack of clarity. In this study, the transcriptomic data we used were derived from 42 farmer cultivars, all of which are from the main cultivated variety Damaya Ginseng in Jilin Province, China. Damaya Ginseng has undergone long-term cultivation and selective breeding in different production areas, resulting in multiple farmer cultivars. The phenotypic differences in saponin content among these cultivars are primarily due to environmental variations in different production regions. Therefore, this transcriptomic data, controlled for the basic genetic background, serves as an ideal resource for exploring the impact of different environmental factors on gene expression and phenotypic variation.
Through the analysis of this transcriptome data, key genes in ginseng, such as OSC, NBS, GRAS, TCP, bHLH, and others, have been successfully identified and analyzed. The relevant findings have been published in the following journals for the reviewer’s reference.
Reference:
[1]. Li Li, Yanfang Wang, Mingzhu Zhao, et al. Integrative transcriptome analysis identifies new oxidosqualene cyclase genes involved in ginsenoside biosynthesis in Jilin ginseng. Genomics. 2021;113 (4):2304-2316. doi: 10.1016/j.ygeno.2021.05.023
[2]. Rui Yin, Mingzhu Zhao, Kangyu Wang, et al. Functional differentiation and spatial-temporal co-expression networks of the NBS-encoding gene family in Jilin ginseng, Panax ginseng C.A. Meyer. PLOS ONE. 2017;12 (7): e0181596-e0181596. doi:10.1371/journal.pone.0181596
[3]. Wang N, Wang K, Li S, et al. Transcriptome-Wide Identification, Evolutionary Analysis, and GA Stress Response of the GRAS Gene Family in Panax ginseng C. A. Meyer. Plants (Basel). 2020;9 (2):null. doi:10.3390/plants9020190
[4]. Liu C, Lv T, Shen Y, et al. Genome-wide identification and integrated analysis of TCP genes controlling ginsenoside biosynthesis in Panax ginseng. BMC Plant Biol. 2024;24 (1):47. doi:10.1186/s12870-024-04729-x.
[5]. Xu J, Chu Y, Liao B, et al. Panax ginseng genome examination for ginsenoside biosynthesis. Gigascience. 2017;6 (11):1-15. doi:10.1093/gigascience/gix093.
[6]. Yang Chu, Shunyuan Xiao, He Su, et al. Genome-wide characterization and analysis of bHLH transcription factors in Panax ginseng. Acta Pharmaceutica Sinica B. 2018;8 (4):666-677. doi:10.1016/j.apsb.2018.04.004.
- Please standardize the formatting of bacterial names.
Authors’ responses:
Thank you for this important reminder. In response, we have conducted a full check of the manuscript to ensure all bacterial names now follow the correct formatting conventions (Line:107-108; 214; 217; 251; 383; 482).
- Prefer using a consistent term, for example, ‘double digestion’ instead of alternating between ‘double digestion’ and ‘double-digested’, to ensure greater consistency.
Authors’ responses:
Thanks. We have thoroughly revised the manuscript to ensure the consistent use of 'double digestion', and we thank the reviewer for their diligence (Line: 215).
- The reviewer suggested changing "activated" to "incubated".​
Authors’ responses:
Thanks. We appreciate the reviewer's insightful comment on the terminology. As suggested, we have replaced the term "activated" with "incubated" in all relevant instances to more accurately describe the experimental procedure and enhance reader understanding (Line: 229).
- Please explain how the plants were sterilized and how the explants were prepared.
Authors’ responses:
We thank the reviewer for this important comment. We have now added detailed information regarding the source of the explant material and the culture process in the 'Materials and Methods' section, which enhances the clarity and completeness of the material preparation steps in our study (Line:109-121).
- The reviewer suggested changing "positive materials" to " positive transformants".​
Authors’ responses:
Thanks. We have revised the manuscript to replace the term "positive materials" with the standard terminology "positive transformants" throughout the text to improve clarity and adhere to academic conventions (Line:267 ;453).
- The reviewer suggested changing "reverse transcribed" to " reverse-transcribed".​
Authors’ responses:
Thanks. In response, we have revised the manuscript by replacing the term "reverse transcribed" with the standardized "reverse-transcribed" throughout the text, ensuring greater clarity and alignment with academic conventions (Line:195; 270).
- The reviewer suggested changing "content" to " quantified".​
Authors’ responses:
Thanks. We have revised the text as suggested, replacing "content" with "quantified" to more accurately describe the analytical action performed (Line:272).
- I recommend that you expand the discussion of your findings by explaining what your results mean in the context of existing knowledge. Please clarify what is already known, whether you have discovered anything new, and how your results compare with previous studies. Additionally, discuss any limitations of your work. It may also be helpful to consider the broader implications of your findings.
Authors’ responses:
Thank you for your constructive feedback. Based on your suggestion, we have expanded and refined the discussion section to provide a more detailed interpretation of our findings in the context of existing research. We have also explored the limitations of our study and discussed the broader implications of our results. We believe these revisions enhance the overall quality, depth, and clarity of the discussion (Line:529-662).
- I suggest that you provide a more detailed explanation or discussion on whether you tried lower bacterial concentrations or alternative methods to reduce toxicity.
Authors’ responses:
Thank you very much for your valuable suggestion. In our VIGS experiments, we did indeed explore multiple approaches, primarily focusing on optimizing the transformation method and timing. Regarding the transformation method, we found that both injection and smearing methods led to wilting of the inoculated leaves, a phenomenon that has been reported in Agrobacterium-mediated transformation of other plants [1,2]. As a result, we explored alternative methods and eventually found that soaking ginseng embryos in Agrobacterium solution allowed for successful embryo development, thus validating the feasibility of the VIGS experiment in ginseng.
In terms of the transformation timing, the unique characteristics of ginseng posed additional challenges. Specifically, the above-ground parts of ginseng (excluding the fruit) are fully formed in the previous year, while the gene silencing phenotype required for VIGS expression needs to manifest on newly grown leaves. Therefore, it was not possible for mature ginseng to produce new leaves with phenotypic changes after VIGS treatment. To overcome this, we conducted the leaf transformation experiment before the flowering stage and observed the white phenotype after the fruiting stage, confirming the success of the VIGS experiment.
We sincerely appreciate the reviewer’s insightful comments. We will expand the discussion section to include this information and provide a more comprehensive explanation of the VIGS technique and its optimization process used in this study (Line:563-604).
Reference:
[1]. Sana Khan, Naveera Fahim, Pooja Singh, et al. Agrobacterium tumefaciens mediated genetic transformation of Ocimum gratissimum: A medicinally important crop. INDUSTRIAL CROPS AND PRODUCTS. 2015;71 (0):138-146. doi:10.1016/j.indcrop.2015.03.080.
[2]. Xin T, Tian H, Ma Y, et al. Targeted creating new mutants with compact plant architecture using CRISPR/Cas9 genome editing by an optimized genetic transformation procedure in cucurbit plants. Hortic Res. 2022;:. doi:10.1093/hr/uhab086
- The conclusion reads more like a summary than a true conclusion. In the conclusion, you should clearly summarize the main findings, the significance of the study, and what it means for future work. Also, highlight the overall contribution of the research to science.
Authors’ responses:
Thanks. We sincerely thank you for your valuable suggestions. Based on your feedback, we have thoroughly revised the conclusion, shifting its focus from a simple summary to a clearer one that includes the main findings, research significance, and other key aspects. The revised conclusion integrates the core findings of the study, elaborates on its specific contributions and scientific significance to the field, and proposes actionable directions for future research. We believe these improvements have strengthened the conclusion and more clearly highlighted the value of this research. Your insightful guidance has been crucial to this revision, and we are deeply grateful for it (Line:664-679).

Reviewer 2 Report
Comments and Suggestions for Authors
Dear Author,
This manuscript addresses a timely and important topic concerning the regulation of ginsenoside biosynthesis in Panax ginseng. The authors successfully identify and functionally characterize the HD-Zip gene PgHDZ01. The work excels both technically and analytically. The primary issue revolves around the precise mechanism explaining the differential accumulation of ginsenoside monomers.
- While the authors confirm a significant positive correlation between PgHDZ01 and Rb1 content, they need to hypothesize and discuss more deeply why the overexpression results show an increase in Rb1 but a simultaneous decrease in other protopanaxadiol-type (PPD: Rd, Rg3) and protopanaxatriol-type (PPT: Rg1, Re) ginsenosides. Does PgHDZ01 regulate specific downstream glycosyltransferases or modification enzymes that prioritize Rb1 formation over others? The VIGS data showed a reduction in key genes PgDDS and UGT71A27, but the overexpression section lacks comparative expression data for these (or other relevant) key enzymes in the hairy root lines. Inclusion and discussion of this data would significantly enhance the mechanistic conclusion.
- The MeJA induction correlation analysis showed PgHDZ01 was extremely significantly correlated with many ginsenosides (Rb1, Rc, Rb2, Rd, F1, F2, Rh1, TOTAL). However, the overexpression data primarily highlights Rb1 increase alongside decreases in Rd, Rg3, Rg1, and Re. The authors should discuss how the wide correlation under MeJA stress relates to the highly specific differential accumulation observed during stable overexpression. This manuscript is technically excellent and makes a significant contribution to plant secondary metabolism regulation. Addressing the above points will strengthen the mechanistic claims and ensure the paper meets the standards of the journal.
Author Response
Authors’ Responses to Editor and Reviewers
We sincerely thank the editor and reviewers for dedicating a significant amount of time and effort to provide valuable suggestions and feedback for this study. We have carefully reviewed all the comments and suggestions and have provided responses to each reviewer's feedback on the manuscript revision webpage. Additionally, in response to each comment, we have clearly marked the location of every modification in the manuscript with the format "Line: [line number]" at the end of our reply. We hope this precise referencing will facilitate your review process.
Should you need additional information, please do not hesitate to let me know.
With my very best regards.
Chaofan Wang, Ph.D.
Title: Functional Study of PgHDZ01 Gene Involved in the Regulation of Ginsenoside Biosynthesis in Panax ginseng
Manuscript ID: plants-3859842
According to the comments from the editor and reviewers, we have re-uploaded the file "plants-3859842-revisions-1." Additionally, we have also uploaded the file "plants-3859842-revisions-1-with the changes highlighted. In this file, relevant comments regarding the responses to the questions raised by the editor and reviewers have been marked in "Track Changes" mode in Microsoft Word. At the same time, in certain sections where supplementary content was added, we have included the necessary references. The updated references are listed in the References section. In addition to addressing the editor's and reviewers' comments, we also identified some errors during our review of the manuscript and made the necessary corrections. The modified sections are as follows:
- On line 108, information about Agrobacterium tumefaciens ( tumefaciens) GV3101 has been added.
- On line 222, information about Agrobacterium tumefaciens ( tumefaciens) GV3101 has been added.
- On line 398, information about Agrobacterium tumefaciens ( tumefaciens) GV3101 has been added.
- On line 411, "chlorophyll b" has been changed to italicized chlorophyll b.
Editor and Reviewer comments:
Editor comment:
- Please carefully check the accuracy of names and affiliations.
Authors’ responses:
Thanks. We have checked and corrected the formatting of the names and affiliations (Line:3-11).
Reviewer #2 comment- Major Concerns:
- This manuscript addresses a timely and important topic concerning the regulation of ginsenoside biosynthesis in Panax ginseng. The authors successfully identify and functionally characterize the HD-Zip gene PgHDZ01. The work excels both technically and analytically. The primary issue revolves around the precise mechanism explaining the differential accumulation of ginsenoside monomers.
While the authors confirm a significant positive correlation between PgHDZ01 and Rb1 content, they need to hypothesize and discuss more deeply why the overexpression results show an increase in Rb1 but a simultaneous decrease in other protopanaxadiol-type (PPD: Rd, Rg3) and protopanaxatriol-type (PPT: Rg1, Re) ginsenosides. Does PgHDZ01 regulate specific downstream glycosyltransferases or modification enzymes that prioritize Rb1 formation over others?
Authors’ responses:
Thank you very much for your insightful and thought-provoking comment. We wholeheartedly agree with your perspective and appreciate the depth of your suggestion. Indeed, the observation of an increase in Rb1 content alongside a decrease in other PPD (e.g., Rd, Rg3) and PPT (e.g., Rg1, Re) ginsenosides is intriguing and warrants further exploration.
As you rightly pointed out, it is possible that PgHDZ01 could be regulating specific downstream glycosyltransferases or modification enzymes that preferentially direct metabolic flux toward Rb1 formation over other ginsenosides. In response to your suggestion, we have expanded our discussion in the manuscript to explore this possibility further, considering the potential role of PgHDZ01 in modulating enzyme activity and influencing ginsenoside biosynthesis pathways, based on current research progress.
Once again, we are grateful for your valuable feedback, which has greatly enhanced the depth and direction of our discussion (Line:635-662).
The VIGS data showed a reduction in key genes PgDDS and UGT71A27, but the overexpression section lacks comparative expression data for these (or other relevant) key enzymes in the hairy root lines. Inclusion and discussion of this data would significantly enhance the mechanistic conclusion.
Authors’ responses:
Thank you very much for your valuable suggestion. We fully agree with your point that measuring the expression levels of key genes involved in ginsenoside biosynthesis in the positive transformants would indeed help deepen our understanding of the molecular mechanism by which PgHDZ01 regulates ginsenoside synthesis. Currently, we are conducting further research on the regulatory mechanisms of PgHDZ01 in ginsenoside biosynthesis and have performed transcriptome sequencing and analysis on the positive transformants of PgHDZ01. Through genome-wide screening, we are identifying both direct and indirect target genes of PgHDZ01, with the aim of constructing a more complete regulatory network. Based on these developments, we plan to report these findings in a future publication, providing a more comprehensive understanding of PgHDZ01’s role in regulating ginsenoside biosynthesis.
Once again, we greatly appreciate your insightful feedback, and we will continue to refine our research accordingly.
- The MeJA induction correlation analysis showed PgHDZ01 was extremely significantly correlated with many ginsenosides (Rb1, Rc, Rb2, Rd, F1, F2, Rh1, TOTAL). However, the overexpression data primarily highlights Rb1 increase alongside decreases in Rd, Rg3, Rg1, and Re. The authors should discuss how the wide correlation under MeJA stress relates to the highly specific differential accumulation observed during stable overexpression. This manuscript is technically excellent and makes a significant contribution to plant secondary metabolism regulation. Addressing the above points will strengthen the mechanistic claims and ensure the paper meets the standards of the journal.
Authors’ responses:
Thank you very much for your insightful and thoughtful comments. We fully agree with your perspective and sincerely appreciate the profound suggestions you have provided. As you observed, PgHDZ01 shows a significant correlation with various ginsenosides under MeJA induction, but exhibits a more specific differential accumulation pattern during stable overexpression, which indeed warrants further exploration. In response to your suggestion, we have expanded the manuscript to provide a deeper analysis of the differences observed between the MeJA treatment and the overexpression experiments. These revisions explore the underlying reasons for the observed differences and will help to deepen the understanding of the phenotypic changes and their associations under different experimental conditions.
Once again, we sincerely thank you for your valuable feedback, which has greatly enhanced the quality of the manuscript (Line:612-624).

Reviewer 3 Report
Comments and Suggestions for Authors
The article is devoted to studying the role of the PgHDZ01 gene in regulating ginsenoside biosynthesis in Panax ginseng. The work is of high quality, combining modern methods of molecular biology and in-depth statistical analysis. The topic is highly relevant: understanding the regulatory mechanisms of secondary metabolite synthesis in medicinal plants is the basis for selection and metabolic engineering.
It is recommended to revise the text for grammatical, spelling, and stylistic errors. Some sentences are not optimally constructed or lack logical flow—these should be reformulated for greater clarity and scientific rigor.
It is advisable to provide more detailed captions for figures and tables so that the reader understands exactly what is being compared or analyzed. In some cases, the captions lack sufficient information.
The description of the statistical methods used (such as p-value, Spearman correlation) should be expanded in the “Materials and Methods” section, specifying the software and parameters employed for calculations.
All abbreviations (such as VIGS—Virus-Induced Gene Silencing, MeJA—methyl jasmonate) should be spelled out at their first mention in the text.
It is recommended to check the structure of the “Results” and “Discussion” sections—some topic shifts are present within blocks, which complicates reading.
There are a number of comments on the text that require further elaboration by the authors. Here is a line-by-line list of them:
84-102 Table or diagram summarizing the gene identification workflow (correlation, co-expression, SNP, etc.) may help.
89 Ensure all acronyms are defined on first use (e.g., QTL, GWAS), even if they might seem standard.
114 Define all acronyms (e.g., MS medium).
123 Please list the R packages used and provide references where possible.
165-175 Perhaps written in excessive detail!
200 What does activation mean? Does the reader need all these details? Maybe it's worth describing the cloning procedure once so we don't have to come back to it again and again?
252 It would be interesting to see in the text exactly how many genes have p < 0.01.
266 Make the inscriptions legible
276 Same comment
297 Same comment
320 Successfully confirmed by three independent methods! Hurray!
483 Perhaps it would be worth considering combining the corresponding 'columns' of the diagrams into one?
Author Response
Authors’ Responses to Editor and Reviewers
We sincerely thank the editor and reviewers for dedicating a significant amount of time and effort to provide valuable suggestions and feedback for this study. We have carefully reviewed all the comments and suggestions and have provided responses to each reviewer's feedback on the manuscript revision webpage. Additionally, in response to each comment, we have clearly marked the location of every modification in the manuscript with the format "Line: [line number]" at the end of our reply. We hope this precise referencing will facilitate your review process.
Should you need additional information, please do not hesitate to let me know.
With my very best regards.
Chaofan Wang, Ph.D.
Title: Functional Study of PgHDZ01 Gene Involved in the Regulation of Ginsenoside Biosynthesis in Panax ginseng
Manuscript ID: plants-3859842
According to the comments from the editor and reviewers, we have re-uploaded the file "plants-3859842-revisions-1." Additionally, we have also uploaded the file "plants-3859842-revisions-1-with the changes highlighted. In this file, relevant comments regarding the responses to the questions raised by the editor and reviewers have been marked in "Track Changes" mode in Microsoft Word. At the same time, in certain sections where supplementary content was added, we have included the necessary references. The updated references are listed in the References section. In addition to addressing the editor's and reviewers' comments, we also identified some errors during our review of the manuscript and made the necessary corrections. The modified sections are as follows:
- On line 108, information about Agrobacterium tumefaciens ( tumefaciens) GV3101 has been added.
- On line 222, information about Agrobacterium tumefaciens ( tumefaciens) GV3101 has been added.
- On line 398, information about Agrobacterium tumefaciens ( tumefaciens) GV3101 has been added.
- On line 411, "chlorophyll b" has been changed to italicized chlorophyll b.
Editor and Reviewer comments:
Editor comment:
- Please carefully check the accuracy of names and affiliations.
Authors’ responses:
Thanks. We have checked and corrected the formatting of the names and affiliations (Line:3-11).
Reviewer #3 comment:
- The article is devoted to studying the role of the PgHDZ01 gene in regulating ginsenoside biosynthesis in Panax ginseng. The work is of high quality, combining modern methods of molecular biology and in-depth statistical analysis. The topic is highly relevant: understanding the regulatory mechanisms of secondary metabolite synthesis in medicinal plants is the basis for selection and metabolic engineering.
It is recommended to revise the text for grammatical, spelling, and stylistic errors. Some sentences are not optimally constructed or lack logical flow—these should be reformulated for greater clarity and scientific rigor.
Authors’ responses:
Thank you for your thoughtful and constructive feedback. We have carefully reviewed and revised the manuscript to address issues related to grammar, spelling, and writing clarity. To ensure the manuscript meets academic journal standards, we also utilized the professional language editing services provided by MDPI, which helped optimize sentence structure, enhance logical flow, and improve overall clarity and scientific rigor. These revisions have significantly improved the manuscript, aligning it with the high standards required by the journal. To confirm the language improvements made, we have attached the English-Editing-Certificate.pdf. We are confident that these revisions have greatly enhanced the readability and academic professionalism of the manuscript.
- It is advisable to provide more detailed captions for figures and tables so that the reader understands exactly what is being compared or analyzed. In some cases, the captions lack sufficient information.
Authors’ responses:
Thanks. We have made supplementary revisions to all the figure and table captions in the manuscript to ensure that readers can more accurately understand the content being compared or analyzed. Thank you for your valuable feedback (Line:294; 315; 339; 358; 364; 427; 447; 479; 511; 517; 522).
- The description of the statistical methods used (such as p-value, Spearman correlation) should be expanded in the “Materials and Methods” section, specifying the software and parameters employed for calculations.
Authors’ responses:
Thanks. We are grateful to the reviewer for pointing this out. As suggested, we have incorporated more detailed descriptions of the statistical methods, software, and relevant parameters into the 'Materials and Methods' section. The specific revisions have been marked in the manuscript (Line:125-128; 141-143; 151-155; 188-193; 238-241).
- All abbreviations (such as VIGS—Virus-Induced Gene Silencing, MeJA—methyl jasmonate) should be spelled out at their first mention in the text.
Authors’ responses:
Thanks. We have carefully reviewed the manuscript and have added the full names of VIGS and MeJA at their first occurrences in the text (Line: 47; 77).
- It is recommended to check the structure of the “Results” and “Discussion” sections—some topic shifts are present within blocks, which complicates reading.
Authors’ responses:
Thank you very much for your valuable suggestion. We sincerely appreciate your thoughtful feedback. In response, we have carefully reviewed and revised the "Results" and "Discussion" sections to improve their structure and clarity, ensuring a smoother logical flow and better readability. We believe these revisions will significantly enhance the overall clarity and depth of the manuscript, aligning it more closely with the journal’s requirements (Line:302-313; 416-417; 529-662).
- Line: 84-102 Table or diagram summarizing the gene identification workflow (correlation, co-expression, SNP, etc.) may help.
Authors’ responses:
Thanks. We have added the following flowchart in the Materials and Methods section, which outlines the process for identifying HD-Zip candidate genes involved in ginsenoside biosynthesis (Line:165-174). The specific modifications and additions are listed below. We hope these revisions will help readers better understand and appreciate the approach used in this study. Thank you again for your valuable feedback.
Table 1. This table summarizes the multi-step screening pipeline for identifying candidate HD-Zip genes involved in ginsenoside biosynthesis. In Step 1, expression–trait correlation analysis was performed using Spearman's correlation in SPSS (p-value ≤ 0.05) to screen genes correlated with ginsenoside content. Step 2 involved three complementary analyses: (2.1) heatmap clustering analysis to select candidate genes that cluster with key genes involved in ginsenoside biosynthesis; (2.2) co-expression network analysis to select candidate genes that interact with key genes in ginsenoside biosynthesis; and (2.3) SNP mutation analysis to select candidate genes significantly associated with changes in ginsenoside content. Step 3 integrated results from all three methods in Step 2 using Venn diagram to select high-confidence candidate genes for experimental validation.
- Line: 89 Ensure all acronyms are defined on first use (e.g., QTL, GWAS), even if they might seem standard.
Line: 114 Define all acronyms (e.g., MS medium).
Authors’ responses:
Thanks. We have thoroughly reviewed the manuscript and have added the full terms for the abbreviations upon their first appearance (Line:101; 116-117; 541-542; 553).
- Line: 123 Please list the R packages used and provide references where possible.
Authors’ responses:
Thanks. We have listed the R package used for generating the heatmap and have also included the relevant references (Line:130-131).
- Line: 165-175 Perhaps written in excessive detail!
Authors’ responses:
Thanks. We thank the reviewer for the valuable comment regarding the level of detail in the Methods section. As suggested, we have revised this section and simplified the description enhance its clarity and conciseness (Line:195-201).
- Line: 200 What does activation mean? Does the reader need all these details? Maybe it's worth describing the cloning procedure once so we don't have to come back to it again and again?
Authors’ responses:
Thanks. We sincerely appreciate your insightful feedback. As suggested, we have revised the term "activation" to "incubated" to enhance precision and improve reader understanding. Additionally, this section describes the establishment of the VIGS system in ginseng. We realized that the previous description was overly detailed, and we have streamlined this part to make it more concise and readable (Line:225-238).
- Line: 252 It would be interesting to see in the text exactly how many genes have p < 0.01.
Authors’ responses:
Thanks. We are grateful to the reviewer for this insightful comment. As suggested, we have revised the Results section to explicitly state the number of genes exhibiting a highly significant correlation (p < 0.01), which provides readers with a clearer understanding of outcomes (Line:278-280).
- Line: 266 Make the inscriptions legible
Authors’ responses:
Thanks. We sincerely thank you for pointing out the issue with the legibility of the inscriptions in our original figure. We have redrawn and enlarged the images as per your suggestion. The revised version is now displayed in the manuscript and has been uploaded to the submission system (Line:292).
- Line: 276 Make the inscriptions legible
Authors’ responses:
Thanks. We sincerely thank you for pointing out the issue with the legibility of the inscriptions in our original figure. We have redrawn and enlarged the images as per your suggestion. The revised version is now displayed in the manuscript and has been uploaded to the submission system (Line:293).
- Line: 297 Make the inscriptions legible
Authors’ responses:
Thanks. We sincerely thank you for pointing out the issue with the legibility of the inscriptions in our original figure. We have redrawn and enlarged the images as per your suggestion. The revised version is now displayed in the manuscript and has been uploaded to the submission system (Line:314-315).
- Line: 320 Successfully confirmed by three independent methods! Hurray!
Authors’ responses:
Thanks. We are grateful for the reviewer's positive feedback and their recognition of the robust cross-verification provided by our integrated approach using heatmap, network, and SNP association analyses.
- Line: 483 Perhaps it would be worth considering combining the corresponding 'columns' of the diagrams into one?
Authors’ responses:
Thank you for your thoughtful suggestion aimed at enhancing the clarity of the figures. We sincerely appreciate your feedback and have given this proposal careful consideration. Before creating the figure, we also contemplated the possibility of combining the corresponding "columns" as you suggested. However, the primary goal of this figure is to clearly demonstrate the broad and significant impact of the overexpression of the target gene on the biosynthesis of various ginsenosides. To most directly support this conclusion, we believe that presenting the data for each ginsenoside in separate panels is the most effective approach. This layout allows readers to easily distinguish the effects of the gene on different ginsenosides (e.g., Rg1, Rb1, Rd), avoiding visual complexity that could arise from combining multiple ginsenoside data into one panel. Therefore, we feel that the original figure layout is better suited to illustrating the gene's effects on ginsenoside biosynthesis.
Nonetheless, we truly appreciate your valuable input on this aspect of the manuscript. Your suggestion has contributed to improving the overall quality of the paper, and we are grateful for your attention to detail.

Round 2
Reviewer 1 Report
Comments and Suggestions for Authors
Thank you for considering the comments and corrections.